


# Primary marine aerosol physical and chemical emissions during a nutriment enrichment experiment in mesocosms in the Mediterranean Sea

Allison. N. Schwier[1], Karine Sellegri[1,] Sébastien Mas[2], Bruno Charrière[3,7], Jorge Pey[4*], Clémence
Rose[1], Brice Temime-Roussel[4], Jean-Luc Jaffrezo[5], David Parin[2], David Picard[1], Mickael Ribeiro[1],
Greg Roberts[6], Richard Sempéré[7], Nicolas Marchand[4] and Barbara D'Anna[8]

[1]Laboratoire de Météorologie Physique CNRS UMR6016, Observatoire de Physique du Globe de
Clermont-Ferrand, Université Blaise Pascal, 63171 Aubière, France
[2]Centre d'écologie marine expérimentale MEDIMEER, UMS3282 OSU OREME, Université de
Montpellier, CNRS/IRD, Sète, France
[3]Centre de Formation et de Recherche sur les Environnements Méditerranéens CNRS UMR5110,
Université de Perpignan Via Domitia, 66860 Perpignan, France
[4] Aix Marseille Univ., CNRS, LCE, Marseille, France
[5]Univ. Grenoble Alpes, CNRS, LGGE, F-38000 Grenoble, France
[6]Centre National de Recherches Météorologiques (CNRM), Météo-France, Toulouse, France [7]Aix-
Marseille Université, Mediterranean Institute of Oceanography (MIO), CNRS/IRD, Université de
Toulon, UM 110, 13288 Marseille, France
[8]Institut de Recherches Sur la Catalyse et l'Environnement de Lyon CNRS UMR5256, Université
Claude Bernard Lyon 1, 69626 Villeurbanne, France
*Now at : Geological Survey of Spain (IGME), 50006 Zaragoza (Spain)

*Correspondence to*: Karine Sellegri (K.Sellegri@opgc.cnrs.fr)

**Abstract.** While primary marine aerosol (PMA) is an important part of global aerosol total emissions, its chemical composition and physical flux as a function of the biogeochemical properties of the seawater still remain highly uncharacterized due to the multiplicity of physical, chemical and biological parameters that are involved in the emission process. Here, nutrient enriched-mesocosms filled with Mediterranean seawater were studied over a three-week period. PMA generated from the mesocosm waters were characterized in term of chemical composition, size distribution and size segregated cloud condensation nuclei (CCN), as a function of the seawater chlorophyll-a (Chl-a) concentration, pigment composition, virus and bacteria abundances. In all mesocosms (enriched and control mesocosms), we detected an enrichment of calcium and a deficit in chloride in the submicron PMA mass compared to





the literature inorganic composition of the seawater. A positive linear correlation was found between the aerosol number concentration flux and the seawater temperature, that could be specific to the mediteranean seawater temperature and organic content. We found that the artificial phytoplankton bloom did not affect the condensation nuclei (CN) or CCN number concentration, the normalized size distribution, the CCN activation diameter or the organic fraction of the PMA compared to the control mesocosm. However, we observed correlations between the organic fraction of the Aitken mode particles and heterotrophic flagellates, viruses, and dissolved organic carbon (DOC). No correlation between the particle organic fraction and Chl-a was observed, contrary to previous observations in natural bloom mesocosm experiments. We believe that this could be due to (1) different complexities compared to natural bloom systems, or (2) longer time periods needed to observe correlation trends.

## 1 Introduction

With global submicron emissions of ~ $10\pm5$ Tg yr$^{-1}$ estimated from modeling studies (Gantt and Meskhidze, 2013), primary marine aerosol (PMA), composed of both sea salt and organic material, is a large and important component of particulate matter in the atmosphere. To fully understand its importance for the climate, the physical flux and composition of the marine aerosol must be well quantified. The aerosol flux properties are highly dependent on multiple physical and chemical properties of the seawater.

The current estimates of PMA emissions can vary drastically depending on how the aerosol flux is parameterized. Size segregated sea spray source functions used in modeling studies include physical parameters such as sea-surface water temperature, salinity, and wind speed (Grythe et al., 2014). Newer sea spray source functions have begun to use the wave roughness Reynolds number, that takes into account the wave development state and implicitly incorporates water temperature and salinity through a viscosity term (Norris et al. 2013; Ovadnevaite et al., 2014; Partanen et al., 2014). However, it has been known for many years that sea spray aerosol also incorporates a significant fraction of organic matter (Blanchard, 1964) which can impact aerosol physical and chemical parameters. The most common method of parameterizing the organic fraction in PMA is to correlate this organic fraction to the surface water concentration of chlorophyll-a. Multiple studies have observed this positive





correlation through such methods as ambient aerosol measurements and chlorophyll-a levels detected by satellite (Ceburnis et al., 2014; O'Dowd et al., 2004; Rinaldi et al., 2013), by modeling the sea spray flux and using satellite chlorophyll-a levels (O'Dowd et al., 2008), and recently by direct PMA generation and simultaneous seawater chlorophyll-a measurements (Schwier et al., 2015). Rinaldi et al.

(2013) observed the strongest correlation using satellite measurements of chlorophyll-a concentrations with a time lag of 8 days between the particulate organic fraction and satellite chlorophyll-a concentration. It is important to note that *in-situ* measurements of ambient marine aerosol include both primary and secondary aerosol formation, while marine aerosol source functions focus solely on PMA formation. Modeling studies have included marine aerosol organic fraction parameterizations to

estimate aerosol emission fluxes (Albert et al., 2012; Vignati et al., 2010) and cloud condensation nuclei (CCN) concentrations (Westervelt et al., 2012). Other studies have observed correlations between the aerosol organic fraction and heterotrophic bacteria abundance (Prather et al., 2013; Schwier et al., 2015), transparent exopolymer particle abundance (TEPs) (Schwier et al., 2015) or dimethyl-sulfide concentrations (Bates et al., 2012). On the other hand, Ceburnis et al., (2014) and Rinaldi et al., (2013)

observed an anti-correlation between aerosol organic fraction and wind speed at 10m above the sea surface at Mace Head that could indicate a significant contribution of secondary organic sources to the ambient aerosol organic content.

The organic fraction of PMA affects the aerosol size distribution or its lognormal mode fitting (Fuentes et al., 2010b; King et al., 2012; Schwier et al., 2015; Sellegri et al., 2006). In some modified seawater

experiments, the size distributions were fit with a lognormal modal distribution and the number fraction of each mode was determined; the number concentration of the Aitken mode increased with changing organic concentrations without affecting the overall size distributions, indicating a change in the type of particles dominating the number size distribution (Collins et al., 2013). The organic fraction has also some impacts on total number concentrations (Fuentes et al., 2010b), as well as the CCN ability (Collins

et al., 2013; Fuentes et al., 2011; Prather et al., 2013). Conversely, other works have found no noticeable effect of marine organic material on CCN activity (King et al., 2012; Moore et al., 2011).

Mesocosm experiments are used to study in-situ marine aerosol production based on constrained water parameters. Prather et al., (2013) and Collins et al., (2013) performed wave channel experiments on





seawater by varying bacteria, phytoplankton and chlorophyll-a concentrations. They observed the largest change of the aerosol activation diameter and the hygroscopicity parameter during the period of augmented heterotrophic bacteria; however these changes did not affect the size distribution of the marine aerosol. Schwier et al. (2015) performed bubble-bursting experiments on acidified mesocosm

water during pre-bloom and non-bloom (oligotrophic) conditions, and found that acidification had no strong effect on physical or chemical parameters. However, pre-bloom conditions enhanced the organic fraction over non-bloom conditions, and also increased the number fraction of the Aitken mode ($D_p$~37.5nm) in the aerosol size distributions. However, this last work that indicates that a direct link between the primary marine aerosol organic fraction and the seawater Chl-a is based on the data sets

issued from two different locations of the Mediterranean Sea at different seasons, and needs to be confirmed with more measurements.

Different marine environments can produce drastic seasonal differences in the aerosol flux and organic concentrations. The Mediterranean Sea is a highly oligotrophic basin (Bosc et al., 2004; MERMEX group, 2011) with seasonal blooms in the northwest of the basin (D'Ortenzio and Ribera d'Alcalà, 2009;

Siokou-Frangou et al., 2010) and high UV-B solar radiation in the stratified surface water during summer time (Sempere et al., 2015). Additionally, the warm water temperatures of the Mediterranean Sea (compared to colder locations such as the North Atlantic Ocean) could also impact the PMA flux (Jaeglé et al., 2011; Ovadnevaite et al., 2014). In this study, we performed a mesocosm experiment in the Mediterranean Sea to study an artificial bloom, different from the natural bloom period studied in

the Schwier et al. (2015) study, and track the physical and chemical properties of primary marine aerosol that allows to discuss the parameterization of organic material in PMA provided in earlier works.

## 2 Methods

### 2.1 Measurement site and mesocosm deployment

A mesocosm experiment was performed from May 5 – 23, 2013 in northwestern Mediterranean Sea at the Station de Recherches Sous-marines et Océanographiques (STARESO), located in the Bay of Calvi,





Corsica. The mesocosms have been described previously in the literature (Vidussi et al., 2011). Briefly, three mesocosms (1.2 m diameter, 3 m height) were filled with 2260L of filtered (<1000μm) natural seawater and deployed in the Mediterranean Sea in the station's bay. They were closed with UV-transparent ETFE roofs, except for periods of sampling, which preserved sunlight irradiance of the

mesocosm while preventing external air from entering the mesocosms headspace above the water. In order to test phytoplankton bloom conditions, one mesocosm remained unchanged as a control (A) and the other two mesocosms (B and C) were artificially enriched with nitrates and phosphates, maintaining the Redfield ratio (N:P=16) (Takahashi et al., 1985). Mesocosm B and C were enriched with nutrients on May 5 to additional 3× ($PO_4^{-3}$: 2.7μM, $NO_3^-$: 42.3 μM) and 1× ($PO_4^{-3}$: 0.88 μM, $NO_3^-$: 14.03 μM)

levels, respectively; Mesocosm C was re-enriched with nutrients to 30× ($PO_4^{3-}$: 27 μM, $NO_3^-$: 423 μM) normal levels in the evening of May 12 after sampling for the day.

In each mesocosm unit, the surface water temperature was monitored frequently (every 10 min) using thermistor probes (Campbell Scientific 107) and the average mesocosm surface-water temperature over the entire campaign was 17.7±0.5°C. Water sampling was performed daily every morning (8h00) by

pumping 20 L of surface water (50 cm depth) by hand from each mesocosm into polycarbonate carboys for analysis. The water was stored in large containers indoors and was covered with black bags to ensure no additional photochemical reactions took place after sampling. Non-mesocosm water (outside water) was also sampled every three days for a suite of biogeochemical and primary marine aerosol measurements.

For dissolved organic carbon (DOC) determination, mesocosm samples were directly collected into sub-sampling precombusted glass bottles. Sub-samples were filtrated through precombusted (450°C, 6 hours) GF/F filters and transferred into 10 mL precombusted (450°C, 6 hours) glass ampoules, immediately acidified with 85% $H_3PO_4$ (final pH~2) , flame sealed, and stored at 4°C in the dark. DOC concentration was measured by high-temperature combustion on a Shimadzu TOC 5000 analyzer, as

described in Sohrin and Sempéré (2005). Deep seawater reference samples (provided by D. Hansell, Univ. of Miami) were run daily to check the accuracy of the DOC analysis. Nitrates and phosphates were measured accordingly to Treguer et Le Corre (1975). Chl-a concentrations were determined using





a fluorescence technique based on a methanol extraction procedure (Raimbault et al. 1988). Nitrate and chlorophyll-a concentration time series over the course of the campaign are shown in Figure 1.

For phytoplankton pigment analysis, samples (0.6-1 L) were vacuum filtered (<200 mm Hg), onto precombusted glass-fiber filters (25 mm, 0.7 mm nominal pore size, GF/F, Wathman), stored in liquid

nitrogen and kept at -80°C until analysis. Pigments were extracted in 2,5 mL of 100% methanol and analyzed by high-performance liquid chromatography (HPLC) as described by Wright et al. (1991). Moreover, two 1.6 mL aliquots for microbial plankton analyses were fixed with glutaraldehyde (0.5% final concentration, previously filtered with a 0.02 μm swinex), incubated 15 to 30 min at 4°C, frozen in liquid nitrogen and then stored at -80°C until analysis. The two aliquots were analysed with FacsCanto

II cytometer (3-laser, 8-color (4-2-2), BD Biosciences) equipped with a 20 mW 488 nm coherent sapphire solid state blue laser to evaluate the abundance of heterotrophic bacteria (Lebaron et al., 2001), flagellates abundance (Christaki et al., 2011) and virus (Brussaard, 2004). Finally, the determination of TEPs abundance was performed by microscopic enumeration following Passow and Alldredge (1994). On each slide, 30 images were taken, at random, using a Olympus AX-70 microscope with a 400x

objective and equipped with a colour camera. For each image, all TEPs were counted manually.

## 2.2 Sea spray generation and characterization

The experimental technique has been described previously for similar types of experiments (Schwier et al., 2015). Briefly, we performed dual bubble-bursting experiments by splashing mesocosm water through plunging water jets, separated into 8 jets, to generate aerosol. Two glass square tanks were

filled with 3.6 L of seawater each (water depth of 10cm), sealed with identical stainless steel lids and flushed with particle-free air (13.7 LPM) through a j-shaped tube to mimic the wind-blowing effect on bubble-bursting. Water was re-circulated via a peristaltic pump at a flow rate of 1.8 LPM. The first ten minutes of every experiment was used as a blank measurement to verify that the aerosol concentration was zero before starting bubbling. All the experimental conditions used (water flow rates, plunging

water depth, etc.) were chosen to follow the work of (Fuentes et al., 2010a). The temperature of the mesocosm water was recorded at the beginning and end of every experiment.




In between each experiment, the tanks and tubing were rinsed with ultrapure ELGA water for 10-15 minutes. To ensure no experimental biases, experiments on mesocosm water (A, B and C) were performed in different orders each day. In order to verify that the sampled water was not altered or affected by daily storage, we performed a bubble-bursting experiment on water sampled from one

mesocosm as both the first and last experiment of the day. We observed no significant changes to any of the physical parameters of the water, verifying that the time necessary to perform all the different experiments did not affect the experimental outcome.

Two tanks were used simultaneously to perform experiments throughout the course of the campaign. The aerosol flow from one tank was passed through a diffusion drier and was impacted onto a 3-stage

impactor (Dekati) at 10LPM for ~1 hr, measuring PM 10, 2.5 and 1. Quartz impactor filters also measured the whole PM1 fraction ($D_p$<1000nm). The quartz fiber filters were immediately stored in a refrigerator after sampling and were later extracted with Milli-Q water and analyzed by ion chromatography (IC) for anions ($Cl^-$, $NO_3^-$, $C_2O_4^{2-}$, $SO_4^{2-}$) and cations ($Na^+$, $K^+$, $NH_4^+$, $Mg^{2+}$, $Ca^{2+}$) (Jaffrezo et al, 1998). Analyses for elemental (EC) and organic carbon (OC) were also performed using

the Thermo-Optical Transmission (TOT) method on a Sunset Lab analyzer (Aymoz et al., 2007) following the EUSAAR2 temperature program proposed in Cavalli et al. (2010). The aerosol flow from the other tank was passed through a diffusion drier and a soft x-ray aerosol neutralizer (TSI Model 3088) before entering a differential mobility particle sizer (DMPS) adapted with a detection using simultaneously a Condensation Particle Counter (CPC) and a miniature continuous-flow streamwise

thermal-gradient CCN chamber (CCNc) (Roberts and Nenes, 2005) to determine particle CCN activation properties. The RH of the flow entering all instruments did not exceed 20%.

For the CCNc-DMPS system, aerosol flow passed first through a TSI-type DMA (length 44 cm) selecting particle sizes ranging from 10-400 nm. The aerosol flow was then split between the CCNc and a TSI CPC model 3010. The DMA sheath flow rate was 7.5 LPM and the sample flow rate was 1.35

LPM, with 1 LPM to the CPC and 0.35 LPM to the CCNc. Multiple charges effects were taken into account, as well as CPC efficiency curve and DMA transfer function in a raw data inversion procedure that followed the European recommendations (http://www.actris.eu/). However, the absence of a PM1 impactor in front of the CCNc-DMPS system led to sampling large aerosol particles with multiple





charges in the 200 nm-400 nm size range. Thus, in the present study we use only the data from the 10-200 nm size range. In the CCNc, a total aerosol flow rate of 100 sccm with a sheath-to-aerosol flow ratio of 5 was used. The CCNc tested two different supersaturations by using a temperature gradient of 5° (dT5) and 6° (dT6) in the column. The top temperature of the column varied as the ambient

temperature changed ($T_{top} - T_{amb} = 2°C$). Calibrations were performed with atomized NaCl solutions at the beginning, end and throughout the campaign. The calibration curves were corrected for doubly charged particles by removing a fraction determined with the height of the plateau in calibration curves (Rose et al., 2008). After this, sigmoidal fits were done separately for each of the curves of activated fraction as a function of particle diameter and the obtained activation diameters ($D_{p50}$) were used to

calculate the CCNc supersaturations. First, the activation diameters were corrected for shape using a factors of 1.08. Then, the corresponding CCNc supersaturation was calculated based on the Köhler theory (Köhler, 1936) as presented in Seinfeld and Pandis (1998):

$$Sc = \exp\sqrt{\frac{256 M_w \sigma_w}{27 R T \rho_w} \left(\frac{\rho_w}{M_w}\right)^3 \left(\frac{\rho_s i_s}{M_s}\right)^{-1} D_{p50}^{-3}}$$

where $M_w$ and $M_s$ are the molecular weights of water (0.018 kg mol−1 ) and solute (0.058 kg mol−1), $\rho_w$ and $\rho_s$ are the densities of water (997 kgm−3 ) and solute (2165 kgm−3 ), R is the gas constant (8.314 J mol−1 K−1 ), T is the temperature (298 K), $\sigma_w$ is the surface tension of water (0.072 Jm−2 ), $D_{p50}$ is the activation diameter and $i_s$ is the van't Hoff factor (Young and Warren, 1992). The van't Hoff factor was estimated to be 2, following the literature recommendations (e.g. Rose et al., 2008). The

temperature differences of 6 ◦ C and 5 °C were deduced to correspond to a supersaturation (SS) of 0.30% and 0.15% respectively.

## 3 Results

### 3.1 Size distributions

The sub-micron size distributions of the aerosol produced from the three mesocosms were very stable

over the entire course of the campaign and were directly comparable to the non-modified outside water




samples. The average size distributions of the three mesocosms combined and the outside water for the entire campaign were fit with 3 lognormal modes (Figure 2 & Table 1). Modal diameters of 25.0±1.5, 49.4±1.7, and 105.4±1.4 nm were observed in the mesocosm samples and similar sizes were found in the outside water samples (24.3±0. 75, 48.7±0.5, and 104.8±0.4 ). The size distributions were then

normalized with respect to the maximum total number concentration and the number fractions of each lognormal mode were calculated. No significant differences were observed temporally in the control mesocosm or the enriched mesocosms, even after the second enrichment in mesocosm C (Figure 3). The average number fractions of Modes 2 and 3 were similar for the three mesocosms (0.41±0.03 and 0.45±0.04, respectively) throughout the course of the campaign, while the number fraction of Mode 1

was smaller (0.14±0.03).

### 3.2 Number and CCN concentrations

The total particle number (condensation nuclei, CN) concentration did not show different features from one mesocosm to the other over the course of the campaign (Table 2). The measured water temperatures increased throughout the course of a one-hour experiment, with an average temperature increase of

5.12±1.8°C hr$^{-1}$. We use the average temperature recorded over the duration of a given experiment. Larger temperatures are systematically observed for the experiments for which the lower CCN supersaturations (SS=0.15%) were investigated, because these experiments were always performed after the SS=30% experiments during which the seawater temperature had increased. We converted our total number concentrations into fluxes (number of particles produced per second per water surface area with

bubbles) using the air flux in the tank (13.7 LPM) and the surface area covered by bubbles, following Fuentes et al. (2010a) and Fuentes et al. (2011) and reported to a whitecap coverage corresponding to a 9 m s$^{-1}$ wind speed (i.e. whitecap coverage of 6.89*10$^{-3}$). No correlation was found between the CN flux and the seawater biogeochemical characteristics. However, we see a significant positive linear correlation ($R^2$=0.35, n=76, p (95% sig) <0.00001) between the total number flux and the experimental

average water temperature (Figure 4):

$$\text{CN Number Flux [s}^{-1}\text{ m}^{-2}] = aT[°C] + b$$

$$\left[a = 2.99 \times 10^4 \pm 4.8 \times 10^3; b = -3.43 \times 10^5 \pm 1.21 \times 10^5\right]$$

(1)



Mårtensson et al. (2003) also observed a positive linear parameterization for water temperature and particle number concentration in the particle diameter range 1.4-1.6 μm, but an anti-correlation for smaller particles (0.0316-0.038 μm). We also explicitly size segregated the flux (Figure 5), which was observed to match previously observed experimental fluxes also shown in Figure 5 (Clarke et al., 2006; Fuentes et al., 2010b; Mårtensson et al., 2003).

The nutrient enrichment of the mesocosms did not affect the CCN activity. The ratio of CCN to condensation nuclei at SS=0.30% (0.15%) remains similar for all three mesocosms over time with CCN/CN$_{average,A}$=0.60±0.03 (0.38±0.03), CCN/CN$_{average,B}$=0.62±0.14 (0.38±0.02), and CCN/CN$_{average,C}$=0.58±0.06 (0.33±0.03) for mesocosms A, B and C respectively, indicating no statistical difference between the control and enriched mesocosms. However, because we observe a correlation between CN and water temperature, this also indicates that a significant correlation ($R^2$=0.22, n=56, p(95% sig)<0.00001) exists between CCN (SS=0.30%) and water temperature, shown below:

$$\text{CCN Number Flux [s}^{-1}\text{ m}^{-2}\text{]} = = aT[°C] + b \tag{2}$$

$$\left[ a = 2.00 \times 10^4 \pm 5.1 \times 10^3; b = -2.51 \times 10^5 \pm 1.24 \times 10^5 \right]$$

### 3.3 Chemical composition, Organic fraction and Activation Diameters

The average inorganic chemical composition of PM1 resulting from impactor sampling is given in Table 3, as the mass fractions of individual components to the total inorganic mass concentration. The average fractions are calculated from 6 individual samples performed on 6 different sampling days per mesocosms (18 samples in total). The inorganic composition of sea spray is very similar from one mesocosm to another, and is very similar to the composition of sea water (Seinfeld and Pandis, 1997; Pilson et al., 1998), with the exception of lower fractions of chloride (on average, 45±5% of the total inorganic mass in sea spray for all mesocosms compared to 55% in typical seawater) and higher concentrations of sodium (on average, 39±3% of the total inorganic mass in sea spray from the mesocosms compared to 30.6% in typical seawater) and calcium (on average, 8±5% of the total inorganic mass in sea spray from the mesocosms compared to 1.2 % in typical seawater). As a result,



the mass ratio of chloride to sodium found in the aerosol phase is on average 1.12±0.17 (from all mesocosms), which is substantially different from the ratio of chloride to sodium reported for seawater (1.8 in Seinfeld and Pandis (1997), Pilson et al. (1998) and Swartz et al. (2004)). The depletion of chloride compared to sodium observed in the aerosol phase, compared to the seawater composition,

cannot be explained by a different relative importance of ions in the Mediterranean coastal seawater compared to literature values, as this relative abundance can be considered constant for all oceans (Dittmar, 1884). The composition of coastal Mediterranean seawater do not deviate from this universal composition for the major ions reported here (Ramzy et al., 2015). Thus the explanation for such a depletion measured for primary sea salt emissions can be either that a fractionation of chloride relative

to sodium happens in the course of the aerosol emission due to a different composition of the sea surface microlayer, or that there is already a chloride volatilization process that takes place during the evaporation of the film drops to aerosol residuals. In the literature, chloride-to-sodium ratios measured in the aerosol phase in ambient marine environments have usually been found to be lower than 1.2 (Bardouki et al., 2003; Koulouri et al., 2008), with the chloride loss usually attributed to a loss in the

form of HCl after sodium chloride has reacted with acidic gases such as $HNO_3$ or $H_2SO_4$ during the "aging" process. Our results would call into question past conclusions about sea salt aging during transport in polluted air masses.

A significant enrichment of calcium in the PM1 primary marine aerosol compared to the composition of the seawater has also been reported in earlier studies (Keene et al. 2007; Cochran et al. 2016; Salter et

al. 2016), with an increased enrichment with decreasing aerosol size. The study by Salter et al. (2016) suggests that some calcium ions are clustering close to the sea surface (probably in the form of carbonate ions), and that if calcium is complexing to organic matter at the seawater surface, it is with minor amounts of organic compounds. In our study, higher chloride deficits and calcium enrichments are observed for enriched mesocosms compared to the control mesocosm which suggests that biological

processes may modulate the deficiency/enrichment of these species in the PMA compared to the seawater. A noticeable portion of free calcium in seawater is used off by marine organisms, especially during the blooming period (Saad and Abdel-Moati, 1991). The photosynthesis process in particular causes the removal of $CO_2$ from the seawater causing the precipitation of $CaCO_3$ (Saad and Hussain,





1978). This process may be enhanced at the sea surface when more light is available, explaining gradients in the calcium carbonate clusters and thus in the enrichment of calcium in aerosol particles compared to bulk seawater composition.

The organic fraction of the primary generated aerosol was addressed using two different approaches. The first one is the classical chemical analysis described above. The organic carbon fraction analyzed from the 18 quartz filters collected using impactor sampling for particles $D_p$<1000nm was very similar for all three mesocosms with an average value of 0.25±0.06 of the total aerosol PM1 mass, corresponding to an organic matter (OM) fraction of 0.31±0.07 when using a OM-to-OC conversion
ratio of 1.4 for primary organic aerosol (Turpin and Lim, 2001). Total PM1 fluxes calculated from the chemical analysis of OM and inorganic species was 0.78, 0.58 and 0.64 µg m$^{-2}$ s$^{-1}$ for mesocosms A, B and C respectively. The chemical composition of PMA generated from Mesocosm A, B and C are shown Figure 6.

The second approach is the retrieval of the organic fraction of the aerosol at its activation diameter
(assuming that organics are hydrophobic) from the total aerosol hygroscopicity (in the form of the parameter kappa). Kappa ($\kappa$) values were calculated from the activation diameters following (Asmi et al., 2012):

$$S(D_p) = \frac{D_p^3 - D_{p,50}^3}{D_p^3 - D_{p,50}^3(1-\kappa)} \exp\left(\frac{4\sigma_w M_w}{RT\rho_w D_p}\right)$$

(3)

where $S$ is the supersaturation, $D_p$ is the diameter of the droplet, $D_{p,50}$ is the dry diameter, $R$ is the gas
constant, $T$ is temperature, and $\sigma_w$, $M_w$, and $\rho_w$ are the surface tension, molecular weight and density of water, respectively (Petters and Kreidenweis, 2007). Kappa values were determined by numerical iteration varying $\kappa$ until the maximum of the saturation curve was equal to the saturation used Inside the CCNc. Surface tension ρw was approximated using the surface tension of water. The aerosol hygroscopicity parameter kappa was measured for two different temperature gradients (supersaturation)
in order to investigate the organic fraction of two different particle size ranges. Table 4 shows the total number concentration, activation diameter, kappa and organic fraction individually for all 3 mesocosms and the outside water; the organic fraction calculated from filter samples is also included.





The activation diameters (Figure 7) at SS=0.30% showed little variance between the control and enriched mesocosms ($D_{p,50,mesocosm\ average}$=59.85±3.52nm). Larger particle sizes (i.e. 90 nm particles measured at a lower supersaturation, SS=0.15%) had higher activation diameters ($D_{p,50,mesocosm\ average}$=93.42±5.14nm), again with little variance seen between the different mesocosms. The kappa value at SS=0.30% was $\kappa_{0.30\%}$=0.74±0.12 and at SS=0.15% was $\kappa_{0.15\%}$=0.82±0.14, both very similar to the marine aerosol kappa average, $\kappa_{marine}$=0.72±0.24 (Pringle et al., 2010). Using these kappa values, the organic fraction can then be calculated using

$$\kappa_{total} = \varepsilon_{org}\kappa_{org} + \left(1 - \varepsilon_{org}\right)\kappa_{inorg} \tag{4}$$

where $\kappa_{org}$ and $\kappa_{inorg}$ are the kappa of the organic and inorganic material, respectively and $\varepsilon_{org}$ is the bulk volume fraction of organic material. $\kappa_{inorg}$=1.25 and $\kappa_{org}$=0.006 were chosen based on Collins et al. (2013) for marine aerosol. The average organic matter (OM) fraction for SS=0.30% (representing 60 nm particles) was 0.41±0.09 and for SS=0.15% (representing 90 nm particles), 0.34±0.11. The organic fraction of the 90 nm particles, representative of the accumulation mode particles, is very similar to the OM fraction obtained from the PM1 filter analysis reported above (0.31±0.07). In comparing the organic fractions measured by filter collection and those calculated from the activation diameters, it is clear that the organic concentration decreased with increasing particle size, following the results of previous studies (O'Dowd et al., 2004; Keene et al., 2007; Schwier et al., 2015).

### 3.4 Correlations with biogeochemical parameters

The different biogeochemical parameters, measured during the experiment, were all compared to the aerosol organic fraction and only some of these biogeochemical parameters seem to be connected to the aerosol organic fraction. Indeed, when combining data from all mesocosms, significant correlations (at the 95% level of confidence, Table 4) were seen between the organic fraction from SS=0.30% ($D_p$=60nm) and viruses ($R^2$=0.13, n=54), heterotrophic flagellates ($R^2$=0.121, n=56), DOC ($R^2$=0.086, n=58), and the pigment zeaxanthin representative of cyanobacteria ($R^2$=0.078, n=52). No correlation was observed with chlorophyll-a concentrations ($R^2$=0.022, n=58 Figure 8). Prather et al. (2013) also observed a correlation between organic fraction and heterotrophic flagellates while no correlation existed with chlorophyll-a.




However, strong linear correlations between chlorophyll-a concentrations and marine aerosol organic fractions have been observed in previous studies, both issued from satellite data (O'Dowd et al., 2004; Rinaldi et al., 2013) and from in situ data (Schwier et al., 2015). We used the organic fractions of larger particles from the filter collection (<1000nm) instead of those calculated from the CCNc to determine if stronger correlations could be observed with the measured biogeochemical parameters. When using this organic fraction, the $R^2$ values of the combined mesocosm data increased, but the significance of the correlations decreased due to the lower number of points available (Table 5). For the PM1 organic fraction though, we still observe a significant correlation (only at the 90% level of confidence) with virus abundances ($R^2$=0.166, n=17), while the correlation with heterotrophic flagellates, DOC and zeaxanthin is lost, but the organic fraction of these larger particles correlate with diatoxanthin (diatomes and dinoflagellates $R^2$=0.41, n=12) and carotene (cryptophytes, $R^2$=0.341, n=12) instead.

If we focus on the correlations between different biogeochemical parameters and the organic fraction for individual mesocosms, we see less significant correlations than for the combined mesocosm data, except for pigments. Using the organic fraction from SS=0.30% (representing 60 nm particles), we observed that the only significant correlation at the 95% confidence level in mesocosm C is with heterotrophic flagellates ($R^2$=0.272, n=19, p=0.010) and in mesocosm B with viruses ($R^2$=0.346, n=18, p=0.005). No correlation was found in mesocosm A, which as the control did not experience large variability in biogeochemical properties over the course of the experiment. At the 90% confidence level, we found DOC to be correlated with the organic fraction of 60 nm particles in mesocosm B ($R^2$=0,187, n=19, p=0.064). More pigments were correlated to the organic fraction of 60 nm particles for individual mesocosms than for the combined mesocosm data, especially for mesocosm C (chorophyll-c2, fucoxanthin, diatoxanthin). More pigments were also correlated to the organic fraction of PM1 particles for individual mesocosms than for the combined mesocosm data, especially for mesocosm B (19'hexafucoxanthin, diadinoxanthin, fucoxanthin, diatoxanthin, beta-beta-carotene). Different pigmented species are favored under nutrient conditions specific to each enriched mesocosm which may influence the organic content of aerosol particles in different size ranges. However, the complexity of the system, and relatively low number of data points in each mesocosm hinders our capability of extracting clear and universal relationships with specific biological indicators.



## 4 Discussion

Based on the consistency of the size distributions and modal diameters throughout the course of the campaign, we observed few effects on aerosol physical parameters from the mesocosm enrichments. The modal diameters observed in this work for the number distributions (25, 49, 105nm) are similar to those observed previously using similar bubble production devices. In acidified seawater mesocosm experiments performed in the Mediterranean, (Schwier et al., 2015) observed the same 4 lognormal modal diameters (18.5, 37.5, 91.5, 260nm) during both pre-bloom and non-bloom (oligotrophic) conditions. (Fuentes et al., 2010a) observed 4 modes (14, 48, 124, 334nm) from similar laboratory experiments with artificial seawater. Modal diameters of 15, 45, 125, 340nm were observed with artificial seawater and West African coastal seawater samples (Fuentes et al., 2010b). Wave-channel experiments with modified Pacific coastal seawater showed 3 modes (~90, 220, 1000nm) (Collins et al., 2013). Sellegri et al. (2006) observed 3 modes (45, 110, 300nm) using a weir to bubble synthetic seawater at 23°C. Hultin et al. (2011) observed either two lognormal modes (site: Askö, 86, 180nm) or three lognormal modes (site: Garpen, 93, 193, 577nm) with Baltic seawater. Bubbling systems used in these various studies vary significantly in terms of number of jets and distance between them, height of the jets to the surface of the seawater, and the presence of a blowing air jet above the surface, etc. The variability of the bubbling systems can largely explain the discrepancies between the bubble size distributions within the seawater, and hence the sea spray size distribution. Overall though, most sea spray size distributions show an Aitken mode around 40-50 nm and a small accumulation mode around 100 nm. Similar systems such as the ones used in Sellegri et al. (2006), Fuentes et al. (2010a, and 2010b), and Schwier et al. (2015) show very similar size distributions to the ones reported in the present study, with an additional nucleation mode around 15-25 nm. In the present study, we did not detect the larger accumulation mode at 300 nm, due to the smaller upper size cut used in our DMPS system.

No change in the modal fraction was observed after either nutrient enrichment to the mesocosms. Other studies have shown an increasing number fraction of the Aitken mode with increasing heterotrophic bacteria concentration (Collins et al., 2013) and during pre-bloom periods (Schwier et al., 2015). In Schwier et al. (2015), the increase in the Aitken mode was positively correlated with viruses, heterotrophic prokaryotes, TEPs, chlorophyll-a and other pigments. In the present study, correlations



were significant between the organic fraction of the Aitken mode and heterotrophic flagellates, viruses and with DOC whereas it was not the case with Chl-a, for the combined data set nor for the individual mesocosms. This result is different from previous observations (Schwier et al., 2015), in which a clear correlation was observed between Chl-a and the organic fraction in the Aitken and accumulation modes

suggesting differences in the speciation of organic material between natural and artificial blooms, which are correlated to the particle size.

The organic mass fraction calculated for ~60nm particles in this work (0.41±0.075) in the unperturbed mesocosm A is intermediate from organic fractions observed during a natural Mediterranean pre-bloom period (60nm, 0.64±0.11) and from the same experiments during oligotrophic seawater conditions

(47nm, 0.24±0.14) (Schwier et al., 2015), indicating that only the correlation in mesocosm A between Chl-a and 60 nm particles' organic fraction would be in line with previous observations. The organic fraction measured in PM1 does fall within the observed range of 30-80% organic mass fractions from sea spray aerosol studies (Collins et al., 2013; Facchini et al., 2008; Keene et al., 2007), however marine organic mass fractions as low as 4% (8% volume) have also been observed (Modini et al., 2010) and the

relevant biogeochemical parameters of the seawater influencing the organic fraction of sea spray are still debatable.

No clear impacts of mesocosm enrichment were observed on the total particle concentrations, being consistent with past literature. However, the experimental temperature affected the number concentration. Hultin et al. (2011), Zábori et al. (2012a, b) previously observed that particle number

concentration decreased with increasing water temperatures in the range 10-14°C whereas it remained constant at higher temperatures. Salter et al., (2014) found that particle concentrations at larger sizes (dry diameter >~ 0.3µm) increased slightly at temperatures between 9 and 30°C while smaller particle concentrations (dry diameter > 0.01µm) remained constant. More recently, Salter et al. (2015) confirm the different behavior of particles with diameters larger than 1 µm from the ones for which diameter

was smaller than 1 µm. They also show a rupture in the temperature dependence of submicron PMA, with a strong decrease of their number concentration with increasing temperature for temperatures below 13 °C, and a very moderate decrease of number concentrations with increasing temperature for temperatures above 13°C. These results are different from the ones observed in our study. While our



temperatures are closer to those tested in Salter et al. (2014), their experiments were controlled at temperatures for up to 5 hours, while our experimental temperatures increased over the course of an hour. However, the number concentration throughout the course of an individual experiment most often either remained constant or increased slightly at all sizes with increasing temperature, and the shape of

the size distribution remained the same regardless of the temperature. Also, the experiments performed in Salter et al. (2014) and Salter et al. (2015) are performed with synthetic seawater, and it can not be excluded that organic matter present in the natural seawater influence the temperature dependence of the aerosol number emission flux. Previous studies have incorporated water temperature effects into marine flux parameterizations. As mentioned, Mårtensson et al. (2003) observed a positive linear

relationship between water temperature and 1.4-1.6 μm diameter particles, but an anti-correlation for smaller particles 0.0316-0.038 μm diameter. Jaeglé et al. (2011) empirically derived a $3^{rd}$ order polynomial relationship between particle concentrations and water temperatures, also observing a positive correlation between the parameters. In addition, Jaeglé et al. (2011) observed strong marine aerosol modeling underestimates in warm waters (>25°C) and overestimates in colder waters (<10°C)

when compared to observations, showcasing the need for better marine aerosol flux parameterization. Partanen et al. (2014) and Ovadnevaite et al., (2014) used the Reynolds number to implicitly incorporate the surface water temperature in global models. Partanen et al. (2014) observed that seasonal wind speed was more important than seasonal water temperature changes in determining the sea-spray aerosol flux. In a global modeling study, Ovadnevaite et al. (2014) observed that warmer

water temperatures enhanced particle flux over colder water. The size resolved flux we observed was similar to those found in most previous studies (Clarke et al., 2006; Fuentes et al., 2010b; Mårtensson et al., 2003), although the experimental apparatus tested were different (Martensson: sintered glass filter with synthetic seawater, Clarke: breaking wave ambient measurements, Fuentes: plunging water jet on collected seawater samples).

**4 Conclusions**

A premeditated large enrichment was performed on two out of three mesocosms to force some physical or chemical changes in the primary marine aerosol artificially generated from the mesocosm seawater.




Overall, there were little differences between properties of submicron PMA generated from the seawaters of the control and enriched mesocosms, indicating that artificial blooms did not strongly affect physical or chemical properties of primary marine aerosol in the time frame of our experiments. In the three mesocosms, we detected an enrichment of calcium and a deficit in chloride in the submicron PMA mass compared to the literature inorganic composition of the seawater. We observed a positive linear correlation between the primary aerosol number concentration flux and the experimental water temperatures between 22°C and 30 °C, which could be specific to the warm waters and organic content of the Mediterranean Sea. Correlations were observed between the organic fractions of the primary aerosol determined from the aerosol hygroscopicity (60nm) and biogeochemical parameters such as heterotrophic flagellates and virus abundance, in agreement with recent mesocosm and wave chamber experiments. Schwier et al. (2015) observed clear differences between pre-bloom and non-bloom Mediterranean periods and found strong correlations between biogeochemical parameters and the organic fraction of 50-60nm particles. Previous studies showed a clear modal fraction differences and decreasing hygroscopicity with the addition of heterotrophic bacteria to natural seawater (Collins et al., 2013; Prather et al., 2013). However, we did not find any clear correlation between the Aitken mode or PM1 organic fractions of PMA and seawater Chl-a, contrary to the findings of Schwier et al. (2015). This could indicate for this work that either (1) isolated artificial blooms in the Mediterranean do not contain the naturally complex biological-organic systems required to have a chemical or physical effect on primary marine aerosol when adjusting only marine nutrients or (2) that any changes caused by artificial blooms require longer observation time periods than those used in this study or affect larger aerosol size than 1 micron.

## Acknowledgements

This study was performed with the financial support of the ANR SAM "Sources of marine Aerosol in the Mediterranean atmosphere" (Grant Number: SIMI-5-6 022 04) and used the experimental facilities of the MEDIterranean center for Marine Ecosystem Experimental Research (MEDIMEER), CNRS Institute of Ecology and Environment (InEE). The authors thank the staff of STARESO (Station de Recherche Sous-Marine et Océanographique) for their hospitality and assistance in the field. Authors wish to acknowledge the support of the MISTRAL-CHARMEX and MISTRAL-MERMEX projects and to thank Elise Hatey and the MARBEC Cécile Roques and the MARBEC technical pole for HPLC





analysis and TEPs measurement. We would like to thank Christophe Salmeron and David Pecqueur from the UPMC/CNRS cytometry/imaging platform of the Banyuls Oceanographic Observatory as well as Patrick Raimbault leader of the M I O-PAPB analytical platform. Finally, the authors gratefully acknowledge the MASSALYA instrumental platform (Aix Marseille Université, lce.univ-amu.fr) for the provision of analysis and measurements used in this publication. Data are available upon request.

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



Table 1. Lognormal modal diameter and number fraction for every mode determined from the average number size distribution. Data from Schwier et al. (2015) for the pre-bloom period tested at Bay of Villefranche (BV) and Fuentes et al. (2010a) for artificial seawater are also shown.

|  | Mesocosm Average | | Outside Water | | Schwier et al. (2015 BV) | | Fuentes et al. (2010) | |
|---|---|---|---|---|---|---|---|---|
|  | Dp (nm) | Fraction | Dp (nm) | Fraction | Dp (nm) | Fraction | Dp (nm) | Fraction |
| Mode 1 | 25.0±1.5 | 0.14±0.03 | 24.3±0.75 | 0.12±0.01 | 20 | 0.19 | 14 | 0.38 |
| Mode 2 | 49.4±1.7 | 0.41±0.03 | 48.7±0.5 | 0.40±0.01 | 37 | 0.48 | 48 | 0.32 |
| Mode 3 | 105.4±1.4 | 0.45±0.04 | 104.8±0.4 | 0.48±0.02 | 92 | 0.24 | 124 | 0.17 |
| Mode 4 | -- | -- | -- | -- | 260 | 0.09 | 334 | 0.13 |

Table 2. Average number concentrations, activation diameters, kappa values ($\kappa$) and organic fractions for each mesocosm at both supersaturations tested. Organic fractions from the filter collections (<1000nm) are also shown for all three mesocosms.

|  | Mesocosm | Number Concentration (cm$^{-3}$) | Activation Diameter (nm) | $\kappa$ | OM Fraction |
|---|---|---|---|---|---|
| SS = 0.30% | A | 2590 ±710 | 59.5 ± 2.5 | 0.74 ± 0.09 | 0.41 ± 0.075 |
|  | B | 2425 ±710 | 60.4 ± 3.9 | 0.72 ± 0.13 | 0.43 ± 0.10 |
|  | C | 2580± 840 | 59.5 ± 4.0 | 0.75 ± 0.13 | 0.40 ± 0.10 |
|  | Outside | 2300 ± 967 | 59.5 ± 2.6 | 0.74 ± 0.095 | 0.41 ± 0.08 |
| SS = 0.15% | A | 2930 ± 864 | 93.3 ± 3.9 | 0.82 ± 0.10 | 0.35 ± 0.08 |
|  | B | 3400 ± 630 | 89.9 ± 4.5 | 0.92 ± 0.15 | 0.26 ± 0.12 |
|  | C | 3090 ± 540 | 97.1 ± 4.5 | 0.73 ± 0.10 | 0.42 ± 0.08 |
| Filters | A | -- | -- | -- | 0.38 ± 0.05 |
|  | B | -- | -- | -- | 0.35 ± 0.11 |
|  | C | -- | -- | -- | 0.34 ± 0.11 |





Table 3. Mass fractions of the main inorganic components in sea spray (PM1) to the total inorganic mass measured from impactor samples on PMA generated from 6 samples per mesocosm and mass fraction of the main inorganic ions in seawater reported by (a) Seinfeld and Pandis (1997), (b) Pilson 1998

| Mass fractions | Cl$^-$ | SO4$^{2+}$ | Na$^+$ | K$^+$ | Mg$^{2-}$ | Ca$^{2+}$ | Cl$^-$/Na$^+$ |
|---|---|---|---|---|---|---|---|
| Meso A | 0.47±0.06 | 0.07±0.02 | 0.38±0.03 | 0.00±0.00 | 0.02±0.00 | 0.05±0.05 | 0.92±0.12 |
| Meso B | 0.43±0.06 | 0.07±0.01 | 0.39±0.03 | 0.00±0.00 | 0.02±0.00 | 0.10±0.06 | 0.78±0.15 |
| Meso C | 0.45±0.02 | 0.06±0.01 | 0.39±0.03 | 0.00±0.00 | 0.02±0.00 | 0.08±0.04 | 0.80±0.17 |
| Seawater[a] | 0.55 | 0.08 | 0.31 | 0.01 | 0.04 | 0.01 | 1.8 |
| Seawater[b] | 0.55 | 0.08 | 0.31 | 0.01 | 0.04 | 0.02 | 1.8 |

Table 4. $R^2$ values, number of sample points (n) and p values at the 95% confidence (p) of correlations between different biogeochemical parameters and the particle organic fractions calculated from the
10  CCNc at SS=0.30% (representing 60nm particles) or from a quartz filter collection (representing <1000nm particles) from averaged data of all mesocosms. p values are given for significant correlations at the 95% confidence level and noted with an asterisk (*) when significant at the 90% confidence level.

| | Organic Fraction, Aitken mode | | | Organic Fraction, PM1 | | |
|---|---|---|---|---|---|---|
| | $R^2$ | n | p | $R^2$ | n | p |
| Total Bacteria | 0.003 | 56 | | 0.016 | 19 | |
| Virus | 0.13 | 54 | 0.007 | 0.166 | 17 | 0.103[*] |
| Hetero. Flagellates | 0.121 | 56 | 0.009 | 0.059 | 18 | |
| Cyanobacteria | 0.026 | 56 | | 0.004 | 19 | |
| Pico-eukaryotes | 0.004 | 56 | | 0.016 | 19 | |
| DOC | 0.086 | 58 | 0.025 | 0.05 | 19 | |
| POC | 0.012 | 29 | | 0.024 | 8 | |





| | | | | | | |
|---|---|---|---|---|---|---|
| PON | 0.006 | 29 | | 0.013 | 8 | |
| TEPs | 0.017 | 58 | | 0.015 | 19 | |
| Chlorophyll-a (fluorometer) | 0.022 | 58 | | 0.073 | 19 | |
| Chlorophyll-b | $8.10^{-4}$ | 54 | | 0.09 | 18 | |
| Chlorophyll-c2 | 0.026 | 58 | | 0.077 | 19 | |
| 19'-Butafucoxanthin | 0.010 | 54 | | 0.104 | 18 | |
| 19'-Hexafucoxanthin | 0.011 | 57 | | 0.001 | 19 | |
| Alloxanthin | $1.0^{-4}$ | 51 | | 0.045 | 17 | |
| Diadinoxanthin | 0.028 | 57 | | 0.122 | 19 | |
| Fucoxanthin | 0.013 | 57 | | 0.195 | 18 | 0.065[*] |
| Lutein | 0.121 | 29 | | 0.095 | 8 | |
| Peridinin | 0.002 | 34 | | 0.055 | 9 | |
| Pheophorbid-a | 0.089 | 11 | | 0.029 | 4 | |
| Prasinoxanthin | 0.077 | 35 | | 0.333 | 11 | |
| Diatoxanthin | 0.036 | 37 | | 0.41 | 12 | 0.023 |
| Beta carotene | 0.008 | 47 | | 0.2 | 16 | 0.081[*] |
| Beta epsilon carotene | 0.029 | 38 | | 0.341 | 12 | 0.044 |
| Zeaxanthin | 0.078 | 52 | 0.044 | 0.018 | 17 | |





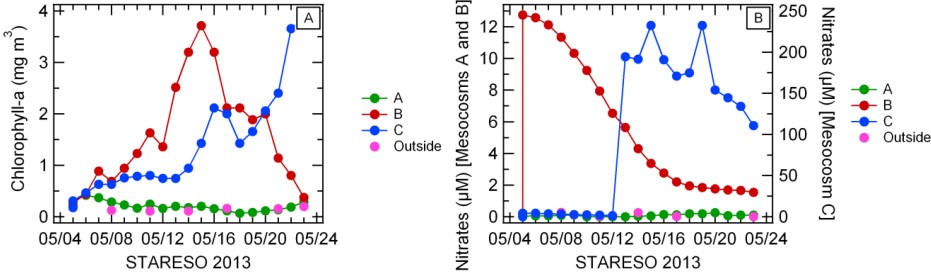

Figure 1. Temporal trends of chlorophyll-a concentrations (Panel 1= P1) measured by fluorometer and

5    nitrate concentrations (Panel 2=P2) measured by Technicon autoanalyzer over the time course of the

experiment. Panel B clearly shows the enrichment of mesocosm B the first day of the campaign and the

enrichment of mesocosm C on May 12.



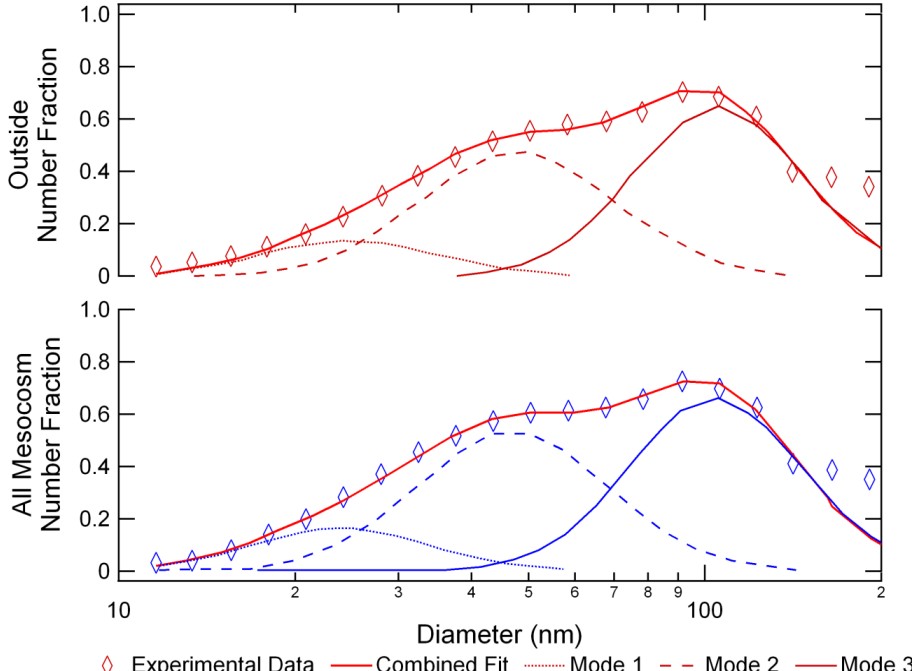

Figure 2. Average size distributions of the outside water and all mesocosms (A, B, and C) at both supersaturations (SS=0.30% and 0.15%) used in this study. The mesocosms are fit with three different lognormal modes.



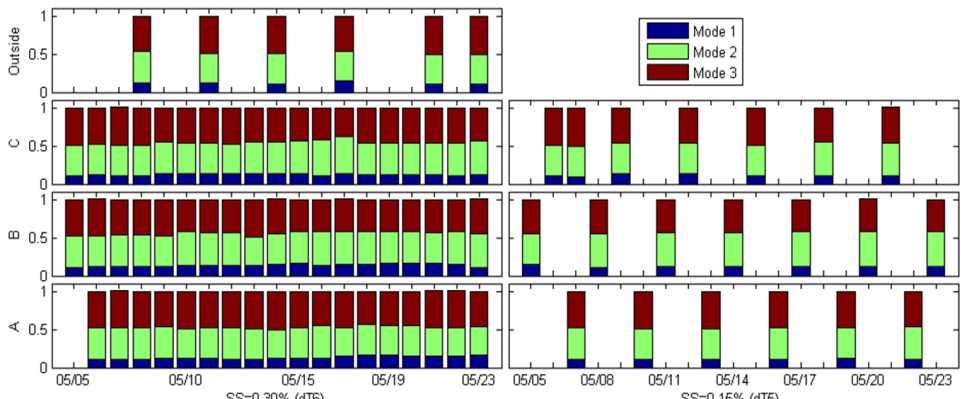

Figure 3. Number fraction of SMPS log normal modes in mesocosms A, B, and C and outside water at SS=0.30% (left) and SS=0.15% (right).

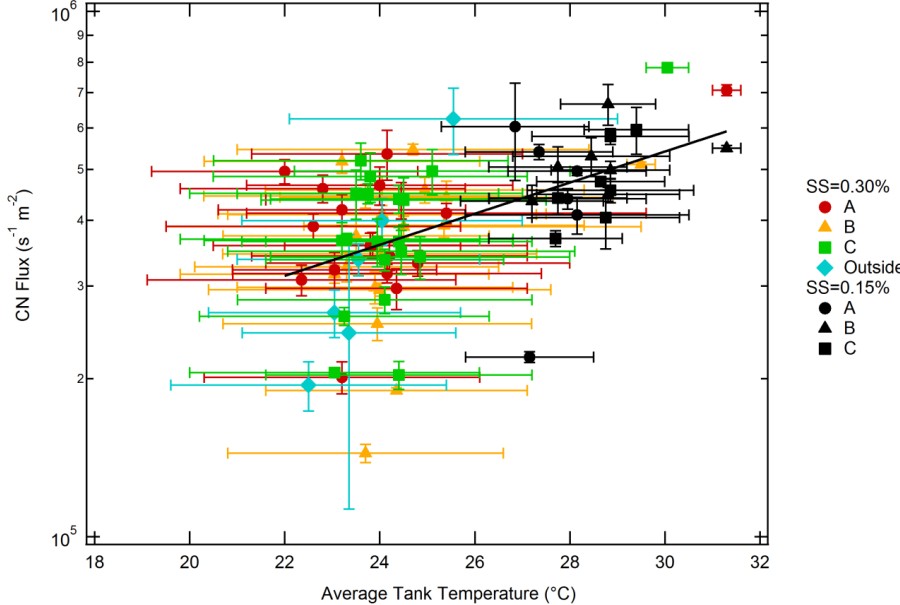

5   Figure 4. Particle flux calculated for a 9 m s$^{-1}$ wind speed vs. average tank temperature for both supersaturations tested.




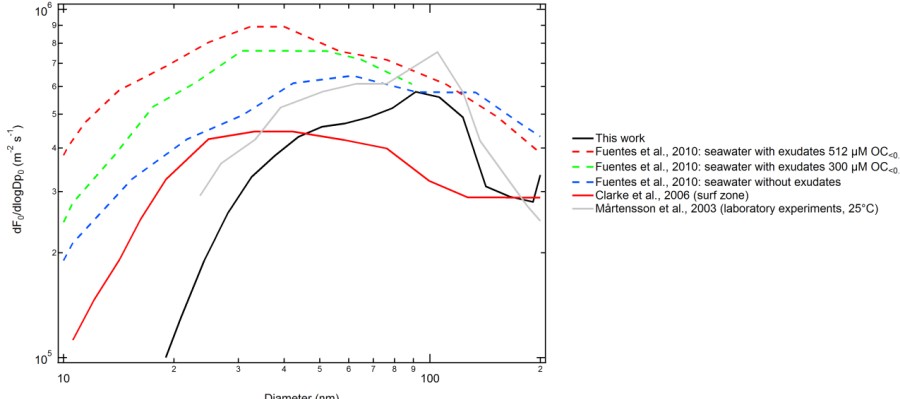

Figure 5. Comparison of size-resolved source flux measurements from previously published work and this work.

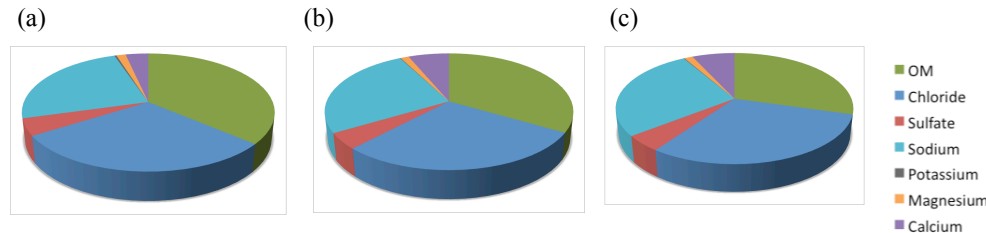




Figure 6. Average chemical composition of the PM1 PMA generated from seawater of (a) Mesocosm A (control), (b) Mesocosm B (enriched) and (c) Mesocosm C (enriched).

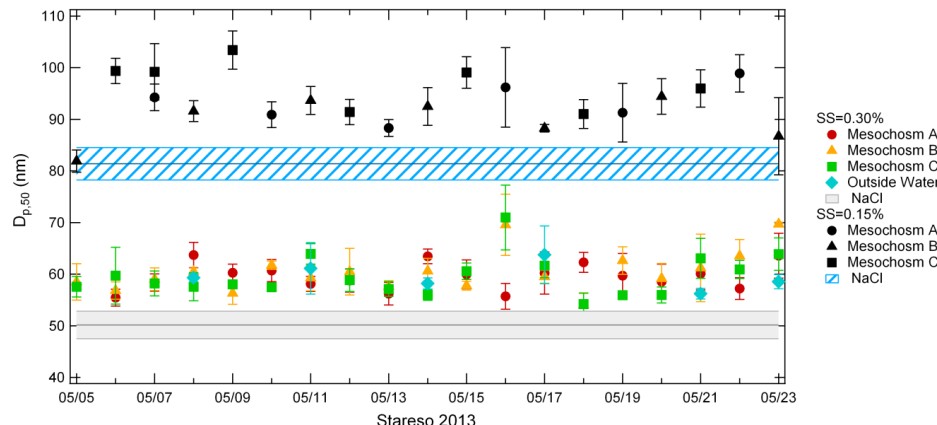

Figure 7. Activation diameters at SS=0.30% and SS=0.15%. The shaded and striped areas indicate the NaCl activation diameters at each given supersaturation as a direct comparison to the seawater samples.





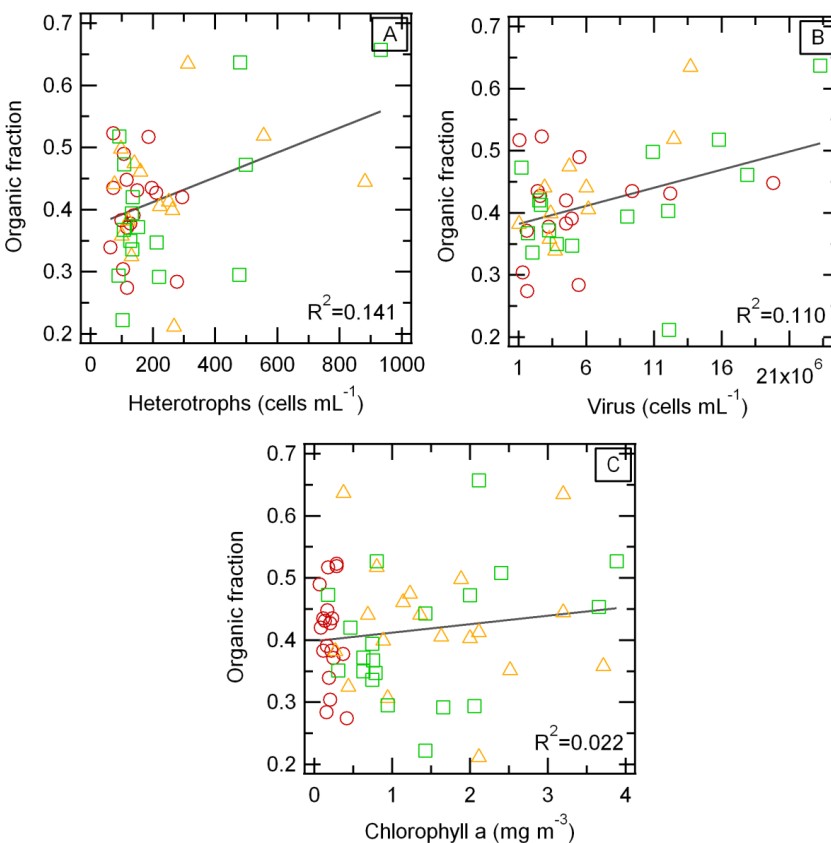

Figure 8. Correlation curves of the organic fraction from SS=0.30% (representative for the Aitken mode particles) versus heterotrophic bacteria abundance (A), virus abundance (B) and chlorophyll-a concentrations (C) with $R^2$ squared values of the fit from all 3 mesocosms combined. Data for mesocosm A (green squares), mesocosm B (yellow triangles), and mesocosm C (red circles) are shown.