# Peer review of "Primary marine aerosol physical flux and chemical composition during a nutrient enrichment experiment in mesocosms in the Mediterranean Sea"

_Atmospheric Chemistry and Physics, 2017_

## Referee Comment (RC1) · Anonymous Referee #2 · 1 Aug 2017

Atmos. Chem. Phys. Discuss., https://doi.org/10.5194/acp-2017-320 Manuscript under review for journal Atmos. Chem. Phys.

Primary marine aerosol physical and chemical emissions during a nutriment enrichment experiment in mesocosms in the Mediterranean Sea

The paper is well written, suitable for ACP, and should be accepted with minor revision.

I found the paper particularly interesting for the effort in the really large array of interdisciplinary measurements made. It is unfortunate that the results are not sticking, but

well worth a publication.

Briefly, I think the paper could be improved by:

- Abstract.

I would number the main conclusion of the paper, the abstract now is too general and I would state what the main results are such as (1) SMPS (2) CCN (3) chemistry (4) correlations (poor) with biology.

- I would stress the correlation with biology (Figure 8) were poor. - I would add a paragraph on the discussion on the time resolution of the filters used (which could not find it on the paper) and if these types of measurements can be useful, or if we need to move to new types of instruments (on line such as ATOFMS, HR AMS, etc) to actually see any valuable variation over time. - I would also stress over the text more the importance of the regionality (point well made) where the Mediterranean oligotrophic waters need more attention, and other mode studied marine aerosol areas may be different.

---

## Referee Comment (RC2) · Anonymous Referee #1 · 1 Aug 2017

The paper of Schwier et al. reports on mesocosm experiments performed in Mediterranean sea. The study is a valuable effort considering the difficulties conducting such experiments and it is unfortunate that no striking results have been found. However, considering scarcity of similar experiments, especially in a natural setting outside the laboratory, the results are worth publishing as they provide useful insights and findings, for example chloride depletion phenomenon or lack of correlation with certain biological parameters. The paper can be accepted after addressing some minor although rather important issues.

[Figure]

Comments:

The title has a typo "nutrient" and awkwardly worded. It should read more like "Primary marine aerosol physical flux and chemical composition during a nutrient enrichment experiment in mesocosms in the Mediterranean Sea".

Abstract is very much like a report of correlations observed or not-observed. More scientific conclusions should be drawn in terms of relationships with biology (looks there was weak, but maybe due to oligotrophic nature) or physics/chemistry. Please summarise conclusions not correlations. Also specify the range of PMA in the abstract.

Page 3, line 25. Ovadnevaite et al., 2011 found dichotomy of HGF and CCN.

Page 4, line 8. Please refer to O'Dowd et al. 2015 on virus mesocosm.

Page 4, line 17. Long et al. 2014 could be an informing study as it was performed in oligotrophic waters.

Page 5, lien 20. ...were directly subsampled into precombusted glass bottles.

Page 6, line 5. Typo 2.5.

Page 6, line 19. "...two square glass tanks".

Page 7, line 9. "impacted onto" to "sampled by".

Page 9, equation 1. The relationship implies zero flux at 11.5C? Something is wrong here.

Page 10, equation 2. Same problem of zero CCN flux at 12.5C.

Page 11. Can authors look into size fraction related chloride depletion? Was it the same in PM10 relative to PM1? Clearly, volatilisation can only occur by chemical reaction which is difficult to reconcile during chamber experiment. The finding calls into question validity of the Na/Cl ratio during aerosol generation. Could it be that due to the smallest particles born from film drops where larger chloride ions are "drained" from

film while in jet drops this does not happen? Calcium enrichment can occur due to entirely different reason, namely being involved in the formation of colloids and gels (with references aplenty).

Page 15, line 22. It is misleading to use nucleation term with respect to sea spray. Possibly "ultra-small mode".

Page 16, line 4. Many of those (non)correlations can be due to chl lag as introduced by Rinaldi et al., 2013 and further explored by O'Dowd et al., 2015.

Table 4. Number of observations is a useful parameter, but P significance takes into account the number of measurements. It can be noted under the Table.

[Figure]

---

## Author Comment (AC1) · 7 Sep 2017

The authors would like to thank both reviewers for careful review of our manuscript and providing us with their comments and suggestion to improve its quality. The following point –by-point responses have been prepared to address all of the reviewers' comments.

Reviewer 1: Abstract. I would number the main conclusion of the paper, the abstract now is too general and I would state what the main results are such as (1) SMPS (2)

[Figure]

CCN (3) chemistry (4) correlations (poor) with biology.

Answer 1: the abstract was re-written as suggested by both reviewers. It now reads:"While primary marine aerosol (PMA) is an important part of global aerosol total emissions, its chemical composition and physical flux as a function of the biogeochemical properties of the seawater still remain highly uncharacterized due to the multiplicity of physical, chemical and biological parameters that are involved in the emission process. Here, 2 nutrient enriched-and 1 control mesocosms filled with Mediterranean seawater were studied over a three-week period. PMA generated from the mesocosm waters were characterized in term of chemical composition, size distribution and size segregated cloud condensation nuclei (CCN), as a function of the seawater chlorophyll-a (Chl-a) concentration, pigment composition, virus and bacteria abundances. The aerosol number size distribution flux was primarily affected by the seawater temperature, and did not vary significantly from one mesocosm to the other. The aerosol number size distribution flux was primarily affected by the seawater temperature and did not vary significantly from one mesocosm to the other. Particle number and CCN aerosol fluxes increase by a factor two when the temperature increases from 22 °C to 32 °C, for all particle submicron sizes. This effect, rarely observed in the literature, could be specific to oligotrophic waters and/or to this temperature range. In all mesocosms (enriched and control mesocosms), we detected an enrichment of calcium (+500%) and a deficit in chloride (-36%) in the submicron PMA mass compared to the literature inorganic composition of the seawater. There are indications that these chloride deficit and calcium enrichment are linked to biological processes, as they are found to be stronger in the enriched mesocosms. This implies a non-linear transfer function between the seawater composition and PMA composition, with complex processes taking place at the interface during the bubble bursting. We found that the artificial phytoplankton bloom did not affect the CCN activation diameter (Dp,50, average=59.85±3.52nm and Dp,50,average=93.42±5.14nm for sursaturations of 0.30% and 0.15% respectively) nor the organic fraction of the submicron PMA (average organic to total mass= 0.31±0.07) compared to the control mesocosm. Contrary to previous observations

in natural bloom mesocosm experiments, the correlation between the particle organic fraction and the seawater Chl-a was poor, indicating that Chl-a is likely not a straightforward proxy for predicting at the daily scale PMA organic fraction in models for all types of sea and ocean waters. Instead, the organic fraction of the Aitken mode particles were more significantly linked to heterotrophic flagellates, viruses, and dissolved organic carbon (DOC). We stress that different conclusions may be obtained in natural (non-enriched) or non-oligotrophic systems. "

Reviewer 2: I would stress the correlation with biology (Figure 8) were poor. - I would add a paragraph on the discussion on the time resolution of the filters used (which could not find it on the paper) and if these types of measurements can be useful, or if we need to move to new types of instruments (on line such as ATOFMS, HR AMS, etc) to actually see any valuable variation over time.

Answer 2: The time resolution of filter measurements is one hour, the same that time resolution of each single bubbling experiment. On-line chemical measurements would require that aerosol is continuously generated with a continuous seawater flow in the bubbling device. This would be of high interest to observe diurnal variations of processes occurring. The conclusions of this paper rather points towards a need for longer time periods of observation rather than to a larger time resolution though.

Reviewer 3: I would also stress over the text more the importance of the regionality (point well made) where the Mediterranean oligotrophic waters need more attention, and other mode studied marine aerosol areas may be different.

Answer 3: We did stress this aspect more clearly throughout the paper, mainly in the discussion part.

---

## Author Comment (AC2) · 7 Sep 2017

The authors would like to thank both Reviewers for careful review of their manuscript and providing with their comments and suggestion to improve its quality. The following point –by-point responses have been prepared to address all of the reviewers' comments.

Reviewer1: The title has a typo "nutrient" and awkwardly worded. It should read more like "Primary marine aerosol physical flux and chemical composition during a nutrient

enrichment experiment in mesocosms in the Mediterranean Sea".

Answer1: Thanks, the title was changed according to suggestions

Reviewer2:Abstract is very much like a report of correlations observed or not-observed. More scientific conclusions should be drawn in terms of relationships with biology (looks there was weak, but maybe due to oligotrophic nature) or physics/chemistry. Please summarise conclusions not correlations. Also specify the range of PMA in the abstract.

Answer2 : The abstract was re-written according to both reviewer's suggestions. The abstract now reads : "While primary marine aerosol (PMA) is an important part of global aerosol total emissions, its chemical composition and physical flux as a function of the biogeochemical properties of the seawater still remain highly uncharacterized due to the multiplicity of physical, chemical and biological parameters that are involved in the emission process. Here, 2 nutrient enriched-and 1 control mesocosms filled with Mediterranean seawater were studied over a three-week period. PMA generated from the mesocosm waters were characterized in term of chemical composition, size distribution and size segregated cloud condensation nuclei (CCN), as a function of the seawater chlorophyll-a (Chl-a) concentration, pigment composition, virus and bacteria abundances. The aerosol number size distribution flux was primarily affected by the seawater temperature and did not vary significantly from one mesocosm to the other. Particle number and CCN aerosol fluxes increase by a factor two when the temperature increases from 22 °C to 32 °C, for all particle submicron sizes. This effect, rarely observed in the literature, could be specific to oligotrophic waters and/or to this temperature range. In all mesocosms (enriched and control mesocosms), we detected an enrichment of calcium (+500%) and a deficit in chloride (-36%) in the submicron PMA mass compared to the literature inorganic composition of the seawater. There are indications that these chloride deficit and calcium enrichment are linked to biological processes, as they are found to be stronger in the enriched mesocosms. This implies a non-linear transfer function between the seawater composition and PMA composition,

with complex processes taking place at the interface during the bubble bursting. We found that the artificial phytoplankton bloom did not affect the CCN activation diameter (Dp,50, average=59.85±3.52nm and Dp,50,average=93.42±5.14nm for sursaturations of 0.30% and 0.15% respectively) nor the organic fraction of the submicron PMA (average organic to total mass= 0.31±0.07) compared to the control mesocosm. Contrary to previous observations in natural bloom mesocosm experiments, the correlation between the particle organic fraction and the seawater Chl-a was poor, indicating that Chl-a is likely not a straightforward proxy for predicting at the daily scale PMA organic fraction in models for all types of sea and ocean waters. Instead, the organic fraction of the Aitken mode particles were more significantly linked to heterotrophic flagellates, viruses, and dissolved organic carbon (DOC). We stress that different conclusions may be obtained in natural (non-enriched) or non-oligotrophic systems.

Reviewe 3: Page 3, line 25. Ovadnevaite et al., 2011 found dichotomy of HGF and CCN.

Answer 3: This reference was added in the text: "Ovadnevaite et al. (2011) found dichotomy of marine aerosol hygroscopicity and CCN, with organics increasing positively the CCN activity of marine aerosols. Âż

Reviewer 4: Page 4, line 8. Please refer to O'Dowd et al. 2015 on virus mesocosm.

Answer 4: We found this study very interesting and rather integrated their results in the section "correlations with biogeochemichal parameters" , sightly modifying our discussion: "The study by Schwier et al. (2015) and the present study were performed with the same experimental set-up, but still show striking opposite results, in term of relationships between the organic fraction of PMA and Chl-a. The main difference between both studies is that in the Schwier et al. (2015) experiments, Chl-a is naturally higher in one mesocosm field campaign that in the other, while in the present experiment, Chl-a was artificially increased via nutrient addition. One hypothesis to explain differences is hence that artificial bloom experiments change biogeochemical equilibrium so that

Chl-a cannot be taken as a proxy for predicting the PMA organic fraction, over a 20 days period scale. In line with this observation, O'Dowd et a. (2015) did not find any relationship between Chl-a concentration and the organic fraction of PMA for laboratory bubble bursting experiments using water cultured in E. huxleyi phytoplankton species. Prather et al. (2013) also observed that no correlation existed with chlorophyll-a, while a correlation was found between organic fraction and heterotrophic flagellates. O'Dowd et al. (2015) and Rinaldi et al. (2013) found a time-lag of 8 days between Chl-a peak concentrations and the organic fraction of marine aerosols in ambient air. It is hypothesized by O'Dowd et al. (2015) that the organic fraction of PMA is linked to the demise of the bloom, due to the release of organic colloids during grazing or viral infection. The correlation found by Schwier et al. (2015) were observed over two separate seasons, which would reinforce the idea that correlation exist at a larger time scale than at the day-to-day basis. However, Schwier et al. (2015) found higher organic fractions of PMA during the pre-bloom period, and not during the period of bloom decay."

Reviewer 5: Page 4, line 17. Long et al. 2014 could be an informing study as it was performed in oligotrophic waters.

Answer 5: Yes, included

Reviewer 6: Page 5, line 20. ...were directly subsampled into precombusted glass bottles.; Page 6, line 5. Typo 2.5.; Page 6, line 19. ". . .two square glass tanks"; Page 7, line 9. "impacted onto" to "sampled by".

Answer 6: All corrected, thanks

Reviewer 7: Page 9, equation 1. The relationship implies zero flux at 11.5C? Something is wrong here; Page 10, equation 2. Same problem of zero CCN flux at 12.5C.

Answer 7: The proposed relationships are only valid for the given range of temperatures. It is likey not libear over the whole temperature range down to 12 $^\circ$C. This now stressed in the text.

Reviewer 8: Page 11. Can authors look into size fraction related chloride depletion? Was it the same in PM10 relative to PM1? Clearly, volatilisation can only occur by chemical reaction which is difficult to reconcile during chamber experiment. The finding calls into question validity of the Na/Cl ratio during aerosol generation. Could it be that due to the smallest particles born from film drops where larger chloride ions are "drained" from film while in jet drops this does not happen? Calcium enrichment can occur due to entirely different reason, namely being involved in the formation of colloids and gels (with references aplenty).

Answer 8: PM10 samples are not accessible right now for ion chromatography analysis, but other yet unpublished data show indeed that PM10 does not show any chloride depletion. The hypothesis of larger chlorine ions drained from film drops is sound, but it would not explain why Ca (40 g mol-1), which is heavier than Chloride, would be enriched. We feel that we can not go into too much speculation without additional measurements at this point.

Reviewer 9: Page 15, line 22. It is misleading to use nucleation term with respect to sea spray. Possibly "ultra-small mode".

Answer 9: yes, corrected

Reviewer 10: Page 16, line 4. Many of those (non) correlations can be due to chl lag as introduced by Rinaldi et al., 2013 and further explored by O'Dowd et al., 2015.

Answer 10: Yes, it could indeed. However, we still found in Schwier et al. (2015) a significant correlation between immediate Chl-a concentrations and the organic fraction of PMA, when comparing pre-bloom conditions over non-bloom conditions. We now discuss this in section 3.4. as previously reported.

Reviewer 11: Table 4. Number of observations is a useful parameter, but P significance takes into account the number of measurements. It can be noted under the Table. Answer 11: Ok, note added.